# Investigation of Mechanical Tests for Hydrogen Embrittlement in Automotive PHS Steels

**Renzo Valentini** [1] , **Michele Maria Tedesco** [2] **, Serena Corsinovi** [3] **, Linda Bacchi** [3] **and Michele Villa** [3,*]

[1]  Department of Civil and Industrial Engineering, Pisa University, 56122 Pisa, Italy
[2]  Metals Department, Centro Ricerche Fiat S.C.p.A, 10135 Turin, Italy
[3]  R&D Department, Letomec S.r.l., 56126 Pisa, Italy
*  Correspondence: m.villa@letomec.com

**Abstract:** The problem of hydrogen embrittlement in ultra-high-strength steels is well known. In this study, slow strain rate, four-point bending, and permeation tests were performed with the aim of characterizing innovative materials with an ultimate tensile strength higher than 1000 MPa. Hydrogen uptake, in the case of automotive components, can take place in many phases of the manufacturing process: during hot stamping, due to the presence of moisture in the furnace atmosphere, high-temperature dissociation giving rise to atomic hydrogen, or also during electrochemical treatments such as cataphoresis. Moreover, possible corrosive phenomena could be a source of hydrogen during an automobile's life. This series of tests was performed here in order to characterize two press-hardened steels (PHS)—USIBOR 1500® and USIBOR 2000®—to establish a correlation between ultimate mechanical properties and critical hydrogen concentration.

**Keywords:** hydrogen embrittlement; ultra-high-strength steels; automotive; press-hardened steels; hydrogen-induced delayed fracture; diffusible hydrogen

---

## 1. Introduction

There is tension between the priority of maintaining and, if possible, increasing the safety of drivers and passengers and the necessity of reducing $CO_2$ emissions and, thus, the weight of vehicles. To resolve this issue, an increasing amount of work is being done on researching and characterizing new high-grade materials which can potentially reduce emissions. Higher mechanical performances are needed to reduce sheet thickness, and serious hydrogen embrittlement susceptibility is the natural consequence.

Advanced high-strength steels (AHSS) exhibit both considerable mechanical properties and good formability. During the production process, hydrogen can be adsorbed in various phases: during electrolytic processes as pickling, electroplating, cataphoresis, and phosphating or even through the moisture present during welding or heat treatment, which can cause hydrogen absorption in the material.

Press-hardened steels (PHS) are hot-formed steels used in automobile structural and safety components, as they present high homogeneity of mechanical properties and excellent fatigue resistance. Their manufacturing process consists mainly of the austenitizing of blanks in an oven, followed by martensitic quenching in a water-cooled stamping tool.

The main source of hydrogen for this kind of material comes from the hot stamping process: atomic hydrogen comes from the dissociation of water at high temperatures, with consequent adsorption and absorption in the steel bulk and its entrapment after water-cooling quenching.

The hydrogen present in the material can diffuse and concentrate in microstructural vacancies or defects, thus cracking the metal lattice. This phenomenon, which is related to hydrogen charging before service, is called internal hydrogen embrittlement (IHE). In this type of attack, the surface is not involved [1].

The most well-known mechanisms that attempt to explain the interaction and behavior of hydrogen [2] with steel are hydrogen-enhanced decohesion (HEDE) and hydrogen-enhanced local plasticity (HELP). The former assumes that hydrogen contributes to the tensile strength: the higher the hydrogen concentration and the higher the hydrogen pressure, the greater the decohesion of steel atoms. The latter suggests that hydrogen, moving towards crack tips where hydrostatic stress is higher, promotes dislocation motions and reduces their interaction energy with internal obstacles, with the consequent formation of microvoids.

Another, more complex mechanism is the so-called adsorption-induced dislocation emission (AIDE). According to this theory, hydrogen adsorption facilitates nucleation of dislocations that can readily and easily move away from the crack tip under the effect of applied stress. The dislocation emission produces crack-advance and crack-opening, resulting in crack growth [3].

In thus study, the traditional testing methods described in the literature to investigate hydrogen embrittlement were evaluated in order to identify which is most suitable for specific applications.

Two different conditions were studied and compared in this work: (1) the slow strain rate test (SSRT), which consists of applying very slow deformation to the material in order to provide hydrogen the time necessary to move and concentrate in the plastic zone; (2) the four-point bending (4 PB) test, which was used to investigate the static behavior of materials.

In addition, a permeation test campaign was performed to evaluate the diffusion coefficient and its dependence on thermal treatment.

The aim of this work was to determine a practical and easily applicable methodology to study the susceptibility to hydrogen embrittlement of ultra-high-strength steels used in the automotive industry.

## 2. Materials and Methods

### 2.1. Materials

The material under investigation was a patented ultra-high-strength aluminum–silicon-coated boron steel which is used in structural and safety automobile components [4]. Two grades were studied: one with an ultimate tensile strength (UTS) of 1500 MPa (USIBOR 1500®) and the other of 2000 MPa (USIBOR 2000®). Their compositions are shown in Table 1.

**Table 1.** Chemical compositions of the two steels (wt %).

| Material | C | Si | Mn | P | S | Al | B | Ti | Cr |
|---|---|---|---|---|---|---|---|---|---|
| USIBOR 1500® | 0.19 | 0.21 | 1.13 | 0.015 | 0.008 | 0.0038 | 0.003 | 0.031 | 0.19 |
| USIBOR 2000® | 0.38 | 0.19 | 1.21 | 0.013 | 0.006 | 0.032 | 0.0032 | 0.024 | 0.28 |

These materials were subjected to a hot-forming process at a temperature between 900 and 950 °C for a period of 240–600 s and then cooled at a rate of about 50 °C/s in the pressing tool to give a martensitic microstructure.

After thermal treatment, the microstructure was completely martensite, due to which its susceptibility to hydrogen embrittlement increased.

The mechanical properties, such as yield strength ($R_{p,02}$), ultimate tensile strength ($R_m$), and elongation at break ($A_{50}$), of the two materials are shown in Table 2.

The difference in the mechanical properties is imputed to the considerable carbon content in USIBOR 2000®.

FEM images were obtained using an SEM FEI Quanta 450 ESEM FEG (FEI, Hillsboro, OR, USA).

**Table 2.** Mechanical properties of the materials under investigation.

| Material | $R_{p,02}$ (MPa) | $R_m$ (MPa) | $A_{50}$ (%) |
|----------|------------------|-------------|--------------|
| 1500 Grade | 1252 | 1485 | 7.3 |
| 2000 Grade | 1510 | 1881 | 6.2 |

## 2.2. Hydrogen Charging

In this study, charging was achieved electrochemically. Samples were immerged in an electrochemical solution and a current between the cathode (the sample) and the anode (a platinum mesh) was imposed to simulate the cataphoresis process. The solutions used contained NaCl and $NH_4SCN$ [5]. The variation and combination of the applied current and the solution's composition, in terms of the recombination's poison ($NH_4SCN$) concentration, allowed for varying the amount of hydrogen absorbed by the material.

In this work, $NH_4SCN$ varied in the range of 0.03–0.3% and the current in the range of 0.25–1 mA/cm$^2$ to avoid surface damage such as cracks at the interface between the coating and the steel due to a high local hydrogen concentration.

Each sample was further treated by placing it in the laboratory furnace and heating it at 150 °C for 10 min, which simulated the paint baking industrial process. In this way, uniform hydrogen distribution was achieved inside the material and, according to [6], not a considerable amount of hydrogen was desorbed.

Figure 1 shows the electrochemical device used for hydrogen charging.

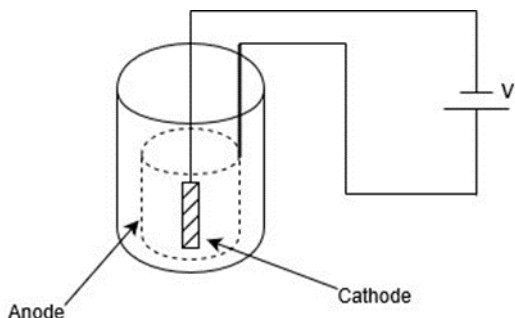

**Figure 1.** Electrochemical cell for hydrogen charging.

## 2.3. Slow Strain Rate Tests

The slow strain rate tests were performed in air, after hydrogen charging, according to ASTM G129 [7]. The crossbar speed was set to 0.001 mm/s and the slow rate corresponded to a very low deformation rate. In this way, hydrogen was able to migrate towards the fracture zone.

The geometrical and dimensional characteristics of the samples used in these tests were in accordance with standards [8] and are shown in Figure 2.

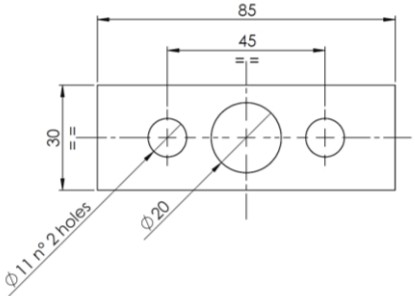

**Figure 2.** Slow strain rate test (SSRT) sample.

### 2.4. Four-Point Bending Tests

The experimental method included 4 PB tests to evaluate the susceptibility of the steel to hydrogen-induced delayed fracture. These tests were carried out according to ASTM F-1624 [9], and the dimensions of the samples are shown in Figure 3.

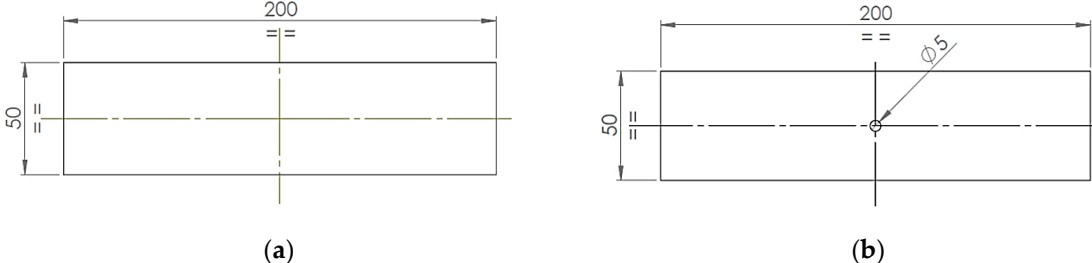

**Figure 3.** Dimensions and geometry of (**a**) four-point bending test samples without a hole and (**b**) with a hole.

The loading procedure consisted of progressive stress that always increased in the elastic range. Starting from 50% of $R_{p,02}$, this load was maintained for 24 h. Then, it was increased every 2 h up to 90% of $R_{p,02}$.

It must be noted that, in case of sample failure, the hydrogen content was immediately evaluated by a hot extraction method to correlate the concentration of hydrogen with the critical load.

Once the 90% of $R_{p,02}$ was reached, this load was maintained for another 24 h, and if the sample did not fail, it was declared "safe".

Figure 4 shows the device used to carry out the four-point bending tests.

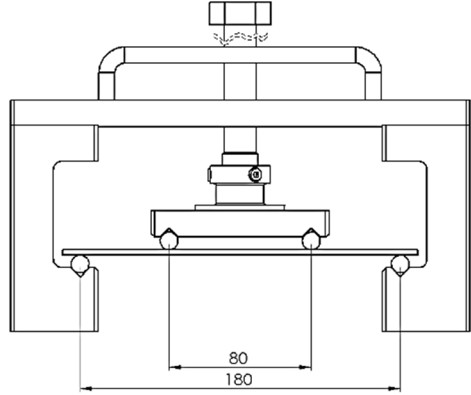

**Figure 4.** Four-point bending test equipment.

### 2.5. Permeation Tests

Permeation tests were carried out according to ISO 17081:2004 [10], with the aim of characterizing materials in terms of the hydrogen diffusion coefficient and by means of an innovative instrument patented by Letomec S.r.l., which uses a solid-state hydrogen-sensitive sensor [11]. The sample was placed between an electrochemical cell and the probe of the instrument and a current between the cathode (sample) and the anode (a platinized titanium mesh) was applied (test solution 3% NaCl with 0.3% $NH_4SCN$ and current density 0.5 mA/cm$^2$). As the discharge of hydrogen occurred on the surface in contact with the solution, hydrogen was then adsorbed and diffused throughout the specimen thickness. Figure 5 shows the scheme of the instrument.

The reaction of hydrogen discharge took place in the sample's surface $M$ in contact with the solution:

$$H_2O + M + e^- \rightarrow MH_{ads} + OH^-. \tag{1}$$

In this way, according to Volmer's theory [12], the atoms of hydrogen could be adsorbed by the material and, in the next moment, be absorbed in the bulk or recombine to produce molecular hydrogen (moving away in the form of gas bubbles).

The absorbed hydrogen reached the exit surface where the sensor's probe was located, and it left the sample passing through the probe and being detected. As a result, the typical permeation curve was obtained.

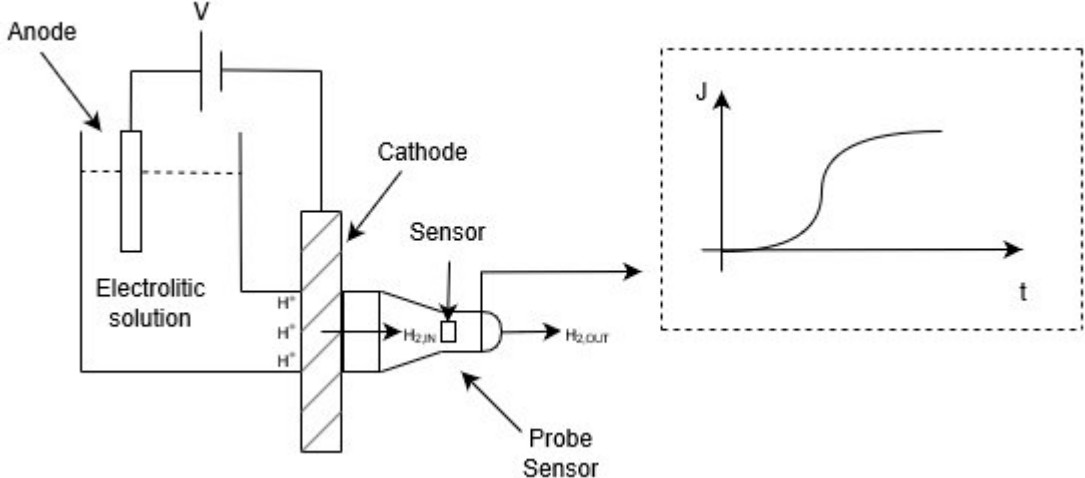

**Figure 5.** Innovative instrument patented by Letomec S.r.l.

## 2.6. Hydrogen Measurement

For 4 PB and SSR tests, the concentration of hydrogen was evaluated by a hot extraction method by means of a Leco DH603 instrument (Leco, St. Joseph, MO, USA). Samples were heated at 265 °C in order to guarantee the complete desorption of diffusible hydrogen [13].

## 3. Results

### 3.1. Slow Strain Rate Tests

The slow strain rate tests were carried out on precharged samples. After sample fracture, the content of hydrogen was measured, as detailed in the previous paragraphs.

Figures 6 and 7 report the characteristics of the two different materials.

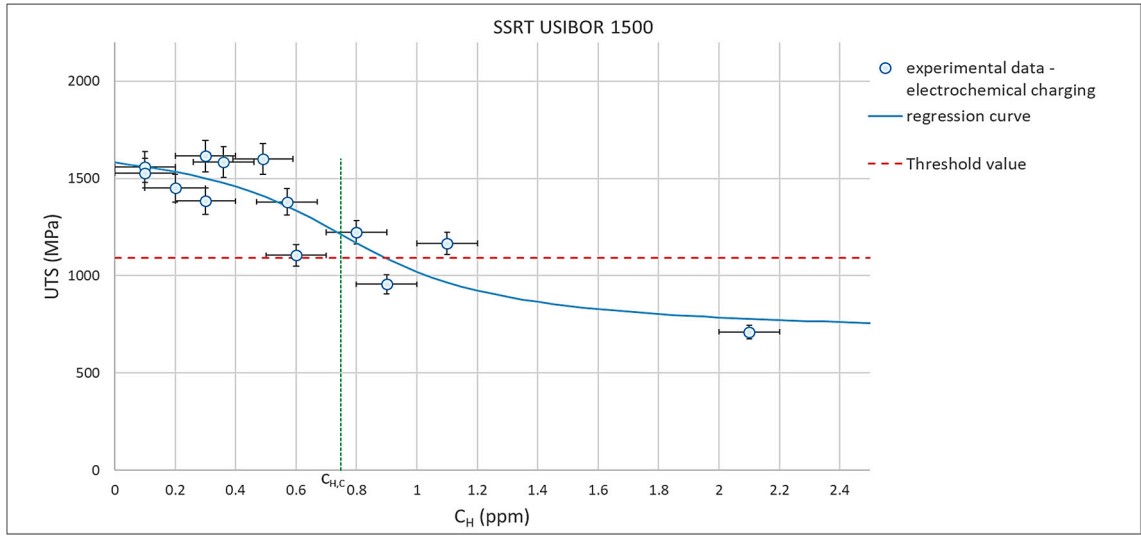

**Figure 6.** SSRT for USIBOR 1500®.

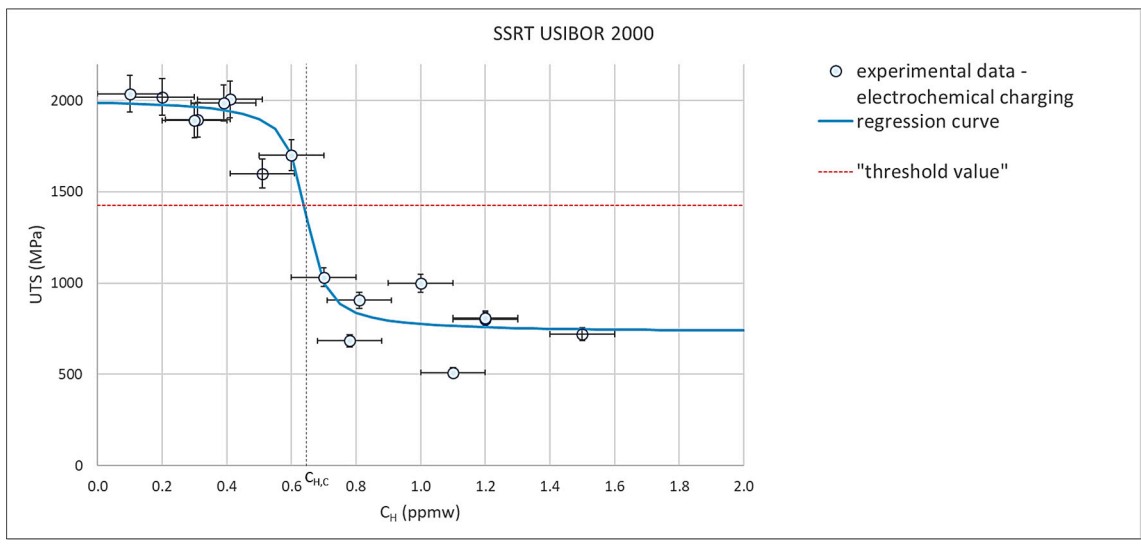

**Figure 7.** SSRT for USIBOR 2000®.

In the figures above, $C_{H,C}$ represents the critical concentration obtained by the inflection point of the regression curve. The threshold value (red dashed line) represents 70% of the ultimate tensile load of the material, and the corresponding hydrogen concentration is the traditional critical value for the examined material. The error bars depend on the sensitivity of the Leco DH603, equal to ±0.1 ppmw, while on the *y*-axis, the error value is 5%, derived from the uncertainty of the measurement of the sample's thickness.

### 3.2. Four-Point Bending Tests

Figure 8 shows the 4 PB test results for USIBOR 1500® and Figure 9 shows those related to USIBOR 2000®. $C_{H,C}$ is again the critical concentration found as the inflection point (and equal to $m_3$ in the regression Equation (7)). As for the SSR tests, the materials presented a first asymptote for low concentrations, a dramatic fall corresponding to the critical concentration, and a second asymptote corresponding to the saturation effect [14].

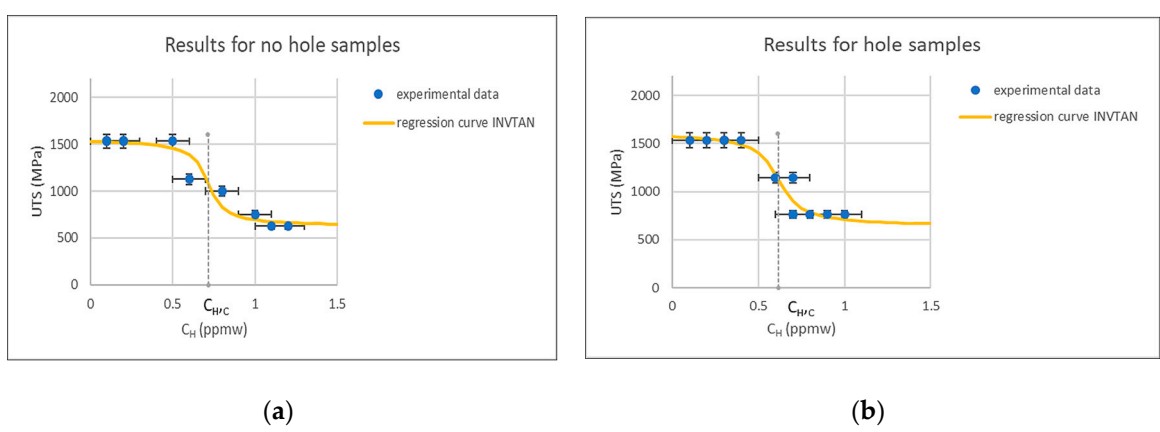

(**a**)　　　　　　　　　　　　　　　　　　　　　　　　(**b**)

**Figure 8.** (**a**) USIBOR 1500® samples without a hole. (**b**) USIBOR 1500® samples with a hole.

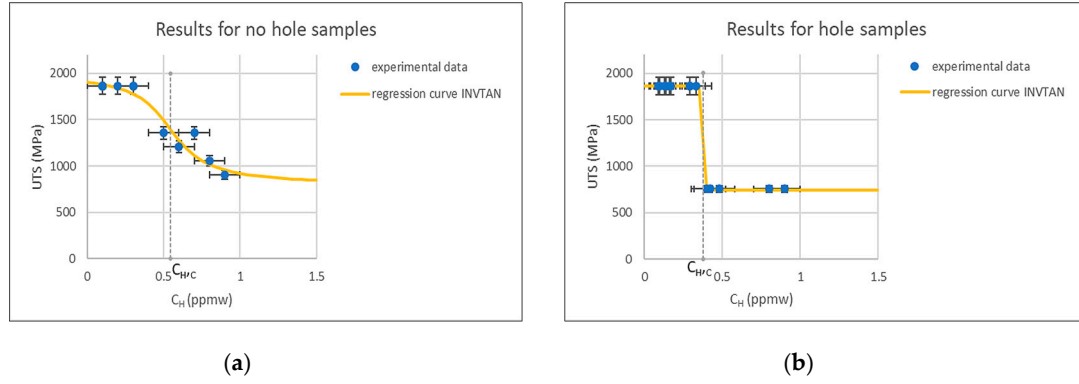

(**a**)                                                                 (**b**)

**Figure 9.** (**a**) USIBOR 2000® samples without a hole. (**b**) USIBOR 2000® samples with a hole.

### 3.3. Permeation Tests

Sample thickness was equal to 1.4 mm in this particular case for both materials. The two steels underwent permeation tests before and after hot stamping treatment to study the effect of microstructure transformation. Figure 10 shows the comparison of the results.

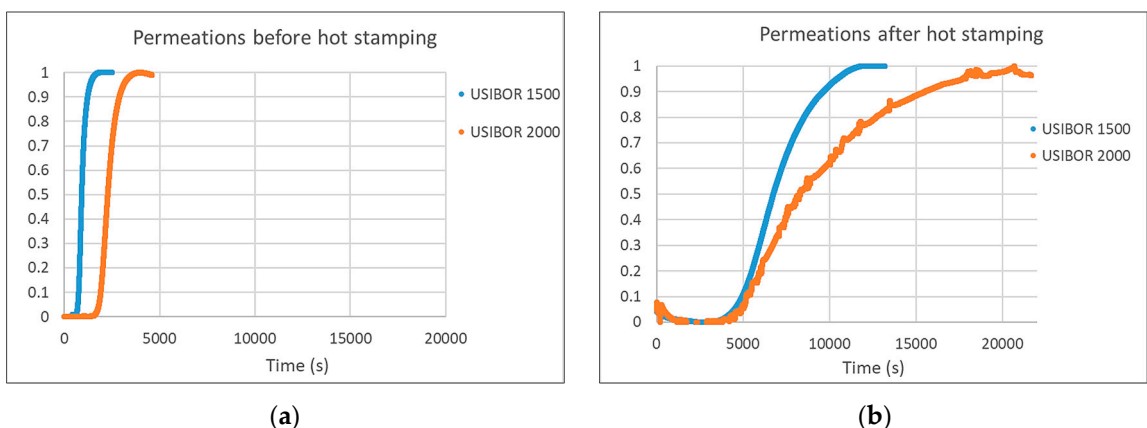

(**a**)                                                                 (**b**)

**Figure 10.** (**a**) Permeation curves before hot stamping. (**b**) Permeation curves after hot stamping.

Table 3 shows the diffusion coefficients for the two different materials under investigation. According to international literature [15], the differences underlined before are evident.

**Table 3.** Diffusion coefficient for the two grades steels as a function of thermal treatment.

| Material | Diffusion Coefficient (m²/s) | |
|---|---|---|
| | **Before Hot Stamping** | **After Hot Stamping** |
| 1500 Grade | $3.39 \times 10^{-10}$ | $4.50 \times 10^{-11}$ |
| 2000 Grade | $1.36 \times 10^{-10}$ | $3.25 \times 10^{-11}$ |

### 3.4. SEM Images

In Figure 11 SEM images are shown, and the typical intergranular fracture surface of SSRT samples is illustrated.

For both pictures, the magnification is 5000× and the hydrogen content absorbed by the samples was 2.1 and 0.5 ppmw, respectively.

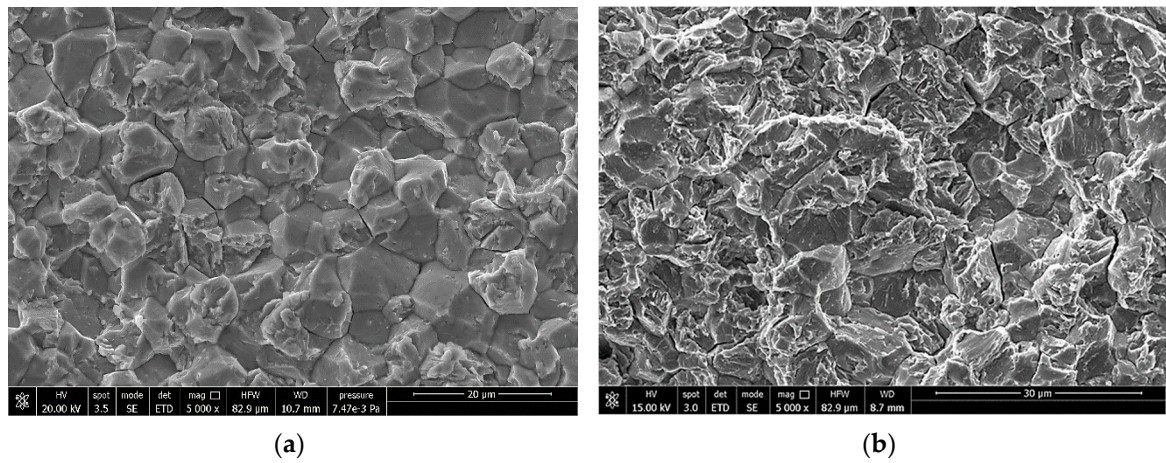

**(a)**　　　　　　　　　　　　　　　　　　　　　　　　　**(b)**

**Figure 11.** (**a**) USIBOR 1500®. (**b**) USIBOR 2000®.

## 4. Discussion

The diffusion phenomenon is regulated by Fick's laws, and even the movement of hydrogen atoms inside steel can be described using the same equations. Nevertheless, the experimental test results demonstrated behavior somewhat distant from the ideal theory. This difference was due to the presence of hydrogen traps inside the metal lattice, such as dislocations, grain boundaries, precipitates, and so on.

Traps influence the diffusion of atoms in the bulk and they are divided in two categories: reversible and irreversible, depending on their own binding energy. Irreversible traps are those that release hydrogen at a temperature higher than 1000 °C, while, according to the literature, reversible traps lose hydrogen at a lower temperature; in particular, for USIBOR, it is equal to 265 °C [13].

According to McNabb et al.'s (1963) model [16], traps saturate and the equations that rule this phenomenon are shown below [15]:

$$\begin{cases} \frac{\partial C}{\partial t} = D\frac{\partial^2 C}{\partial x^2} - N_r\frac{\partial v}{\partial t} - N_i\frac{\partial w}{\partial t} \\ \frac{\partial v}{\partial t} = K_rC(1-v) - pv \\ \frac{\partial w}{\partial t} = K_iC(1-w) \end{cases} \tag{2}$$

where $C$ is the hydrogen concentration (atoms/m³); $D$ is the hydrogen diffusion coefficient (m²/s) in pure iron; $N_r$ and $N_i$ are the concentrations of reversible and irreversible traps, respectively (atoms/m³); $v$ represents the occupied reversible trap fraction, while $w$ refers to irreversible traps; $t$ and $x$ are the time and space variables, respectively; $K_r$ is the trapping rate for reversible traps (m³/atoms s); $K_i$ is the same for irreversible traps; and $p$ is the release rate for reversible traps.

As can be noticed in Table 3 and according to [15], traps are dislocations generated from the phase transformation during the quenching process, and here, the diffusion was always faster in USIBOR 1500® than in USIBOR 2000® because of the considerable amount of carbon in the higher grade; moreover, the thermal treatment reduced the diffusion coefficient of hydrogen.

The embrittling process [17] is related to the interaction of atomic hydrogen with interatomic bonds, and when the cohesive strength of the material is overcome, crack propagation occurs, which refers to the I mode of crack growth according to mechanical fracture. At the crack tip, the presence of a high pressure gradient increases the solubility of hydrogen in the lattice due to lattice expansion, resulting in a hydrogen flux towards this region.

According to the HEDE model, the cohesive strength is affected only by lattice hydrogen $C_L$ and its dependence is shown in the following equation:

$$\sigma_N^c(C_L) = \sigma_N^c\left(1 - a_1\theta_s + a_2\theta_s^2\right) \tag{3}$$

where $a_1$ and $a_2$ are empirical constants and $\theta_s$ is defined as

$$\theta_s = \frac{C_L}{C_L + \exp\left(-\frac{\Delta H}{RT}\right)} \tag{4}$$

where $\Delta H$ is equal to 30 kJ/mol, R is the gas constant, and T is the absolute temperature.

The hydrogen concentration tends to accumulate next to the crack tip because of the higher hydrostatic stresses, according to the well-known Beck's law [18]:

$$C = C_L \exp\left(\frac{V_H \sigma_H}{RT}\right) \tag{5}$$

where $V_H$ is the molar volume of hydrogen in the lattice, equal to $2 \times 10^{-6}$ m³/mol; $\sigma_H$ is the hydrostatic stress; R is the gas constant; T is the absolute temperature; $C_L$ is the concentration of hydrogen without stress; and $C$ is the concentration near the crack tip in the presence of stress [19].

The accumulation of hydrogen in potential cracking initiation sites depends on the strain–stress time gradient and diffusion coefficients [20]:

$$\frac{\partial C}{\partial t} = D\nabla^2 C + D\frac{V_H}{R(T - T^z)}\nabla C\nabla p + D\frac{V_H}{R(T - T^z)}C\nabla^2 p \tag{6}$$

where $C$ is the hydrogen concentration, $D$ is the diffusion coefficient, $T^z$ is the absolute zero temperature, and $p$ is the hydrostatic stress.

In international literature, it is possible to find various studies of numerical and theoretical simulations in order to explain hydrogen embrittlement phenomena [20].

In the automotive industry, very thin sheets ($0.5$–$2 \times 10^{-3}$ m) are used, which is the reason why a more practical approach is reasonable.

The extrapolation of a regression curve (Figures 6 and 7) can describe the behavior of steel [21,22], and the corresponding equation was derived here as

$$\text{UTS (MPa)} = m_1 - m_2 \times \text{arctg}\left(\frac{C_H - m_3}{m_4}\right) \tag{7}$$

From the function's study, the point where the second derivative was equal to zero had abscissa equal to $C_H = m_3$, which is very close to the value of the critical hydrogen concentration, obtained according to [23], for both steels.

Similarly, for the four-point bending tests and from the mathematical expressions, the critical hydrogen concentration was found for both materials, the results of which are summarized in Table 4.

**Table 4.** Critical hydrogen concentrations of the different mechanical tests.

| Material | USIBOR 1500® | USIBOR 2000® |
|---|---|---|
| 4 PB Hole | 0.61 | 0.37 |
| 4 PB No Hole | 0.71 | 0.54 |
| SSRT | 0.74 | 0.64 |

Samples subjected to high stress intensification, due to the hole, required less average hydrogen concentration to reach the local hydrogen accumulation that leads to fracture.

For SSRT specimens, the deformation rate was higher than the diffusion and the hole created an intensity factor close to 1. For this reason, the accumulation of hydrogen in the SSRT sample was negligible. In contrast, the presence of the hole in the 4 PB samples constituted the worst condition. The 4 PB samples without the hole showed a concentration closer to SSRT specimens, despite the difference in test duration, due to the absence of the stress gradient.

Figure 12 shows the evolution of the hydrogen concentration at the crack tip as a function of test time.

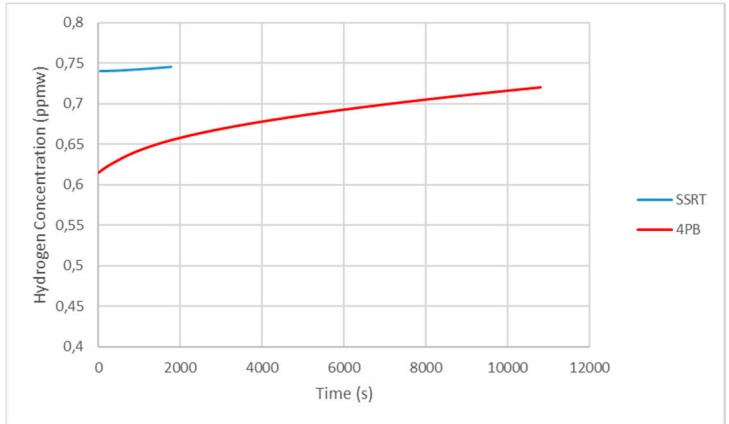

**Figure 12.** Hydrogen accumulation at the crack tip during the test. Note: simulation refers to USIBOR 1500®.

## 5. Conclusions

From the different mechanical tests performed on USIBOR 1500® and USIBOR 2000®, the following conclusions can be derived:

- These steels are sensitive to hydrogen delayed fracture and the grade 2000 is considerably more sensitive than grade 1500.
- The 4 PB tests with the hole were the most severe because the failure occurred with a minimum average concentration.
- The hydrogen diffusion strongly decreased after the hot stamping and quenching process because of the martensite formation; moreover, because of the greater amount of carbon, grade 2000 presented slower diffusion.
- The differences in critical hydrogen concentration values for the mechanical tests were due to two factors:

  1. The effect of the deformation rate in the SSRT, which provided hydrogen with little time to diffuse near the crack tip.
  2. The presence of the hole in the 4 PB samples induced stress gradients that, coupled to time, created hydrogen accumulation near the crack tip.

- Finally, the 4 PB samples with the hole were absolutely the most realistic at simulating the risk of hydrogen embrittlement for this type of material. However, the SSRT is a quick method to compare the behavior of different materials in the presence of hydrogen.

**Author Contributions:** S.C., R.V., and M.V. designed and conceived the experiments; M.V., S.C., and L.B. performed the experiments; all the authors analyzed data; M.M.T. provided the materials; M.V. and R.V. wrote the paper.

**Funding:** This research work was implemented under the umbrella of the Formplanet Project—Sheet Metal Forming Testing Hub, HORIZON 2020 Grant Agreement ID: 814517.

**Acknowledgments:** The authors wish to thank Eng. Barile and Eng. Thierry (Arcelor Mittal) for materials, and Bernardo Monelli and PhD student Francesco Aiello for the FEM simulation.

**Conflicts of Interest:** The authors declare no conflict of interest.

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
