# Peer review of "Investigation of Mechanical Tests for Hydrogen Embrittlement in Automotive PHS Steels"

_metals, doi:10.3390/met9090934_

Round 1
Reviewer 1 Report
The manuscript reported that hydrogen embrittlement properties of hot stamping steels evaluated by SSRT and 4-point bending constant load tests were conducted, and were compared each other. The improvement of hydrogen embrittlement is important mission to apply the hot stamping steel for automotive parts. It is expected that the results support the assessment of hydrogen embrittlement of automotive high strength steels. The comments are as follows.
line 63:
The manuscript compared the effect of microstructure on hydrogen embrittlement properties of the hot stamping steels. Therefore, characterization of microstructure is important information. Authors should indicate hot forming condition, such as austenitizing temperature, time and cooling rate. In addition, SEM image of microstructure before and after hot stamping are needed.
Table 2:
Authors should explain the abbreviations in Table 2 in the caption.
line 86:
Bake hardening process was conducted at 150 ℃ for 10 min. Is it a conventional condition? I felt time was so short.
lines 85-88:
The microstructure of hot stamping steels is usually martensite. Hydrogen in martensite is easy to diffuse at room temperature. Thus, when steels were heated at 150 ℃, hydrogen is desorbed from the sample. Are there any evidence to remain the hydrogen in the hot stamping steels after bake hardening process?
line 96:
What is the strain rate at deformed part in SSRT tests? A crosshead speed of 0.001 mm/s is not slow crosshead speed.
Figures:
It is difficult to confirm Figs. 2, 3, 6, 7, 8, 9 and 10. The illustrations and graphs are very thin, and we can not confirm the values of axes. Authors should indicate clear figures in the manuscript.
line 111:
Is the hydrogen charging time of 2 hours reasonable for homogeneous distribution of hydrogen in the steel? In Fig. 13, hydrogen content contentiously increased until 10000 s. This result suggests that 2 hours is not enough to obtain the homogeneous hydrogen distribution in the steel.
Figure 4:
Would you please show the distances of outer and inner fulcrums? It is important to calculate the stress and strain of 4-point bending specimens.
line 145:
How long did you keep the sample at 265 ℃? If sample was only heated at 265 ℃, is all diffusible hydrogen desorbed from steels?
line 152:
“In Figure 6 and Figure 7 were reported …” should “Figure 6 and Figure 7 report …” There are some same kind of expressions in the manuscript. Please correct them.
Figure 6:
Did authors consider the stress concentration in the vertical axis? The sample exhibits hole at the center.
12: Figure 11:
Scale bar is needed in the fracture surfaces.
13: Discussion, Figure 13:
In the manuscript, the effect of critical hydrogen concentration on hydrogen embrittlement was discussed. However, if applied stress is changed, the critical hydrogen content should be changed. The authors should discuss the hydrogen embrittlement properties with both critical hydrogen content and applied stress.
Author Response
Response to Reviewer 1 Comments
Point 1 (line 63):
The manuscript compared the effect of microstructure on hydrogen
embrittlement properties of the hot stamping steels. Therefore,
characterization of microstructure is important information. Authors should
indicate hot forming condition, such as austenitizing temperature, time and
cooling rate. In addition, SEM image of microstructure before and after hot
stamping are needed.
Response: the information required are added to the paper; SEM images of microstructure before and after hot stamping are not attached because it’s very common to find them in international literature [Valentini, Tedesco, Corsinovi, Bacchi]
Point 2 (Table 2):
Authors should explain the abbreviations in Table 2 in the caption.
Response: Done
Point 3 (line 86):
Bake hardening process was conducted at 150 ℃ for 10 min. Is it a
conventional condition? I felt time was so short.
Response: this is a paint baking simulation process but in this paper it’s useful to avoid a pick of hydrogen in surface; moreover this kind of treatment is really carried out in plant manufacture process. According to [Kim, Park, Lee, Yoo, Jung] time was set in order to avoid total hydrogen release to homogenize the diffusible hydrogen concentration.
Point 4 (lines 85-88):
The microstructure of hot stamping steels is usually martensite. Hydrogen in
martensite is easy to diffuse at room temperature. Thus, when steels were
heated at 150 ℃, hydrogen is desorbed from the sample. Are there any
evidence to remain the hydrogen in the hot stamping steels after bake
hardening process?
Response: In addition to what explained in the previous point (line 86), steel is coated with Al-Si layer and it acts as a barrier for hydrogen, thus hydrogen desorption is really limited.
Point 5 (line 96):
What is the strain rate at deformed part in SSRT tests? A crosshead speed
of 0.001 mm/s is not slow crosshead speed.
Response: the crossbar speed of 0.001 mm/s corresponds to a deformation measured by means of an extensometer of 4.5 E-5 s-1, inside the range recommended by standards.
Point 6 (Figures):
It is difficult to confirm Figs. 2, 3, 6, 7, 8, 9 and 10. The illustrations and
graphs are very thin, and we can not confirm the values of axes. Authors
should indicate clear figures in the manuscript.
Response: Done
Point 7 (line 111):
Is the hydrogen charging time of 2 hours reasonable for homogeneous
distribution of hydrogen in the steel? In Fig. 13, hydrogen content
contentiously increased until 10000 s. This result suggests that 2 hours is not
enough to obtain the homogeneous hydrogen distribution in the steel.
Response: See response to point 3; the diffusible hydrogen content after 2h is very close to homogeneous condition.
However, as said before, paint baking simulation treatment is useful also to reach an homogeneous distribution.
Point 8 (Figure 4):
Would you please show the distances of outer and inner fulcrums? It is
important to calculate the stress and strain of 4-point bending specimens.
Response: done
Point 9 (line 145):
How long did you keep the sample at 265 ℃? If sample was only heated at
265 ℃, is all diffusible hydrogen desorbed from steels?
Response: the measuring time depends on the sample under investigation: however, test is interrupted only once the hydrogen signal returns to zero.
Point 10 (line 152):
“In Figure 6 and Figure 7 were reported …” should “Figure 6 and Figure 7
report …” There are some same kind of expressions in the manuscript.
Please correct them.
Response: Done
Point 11 (Figure 6):
Did authors consider the stress concentration in the vertical axis? The
sample exhibits hole at the center.
Response: The stress concentration factor in the case of SSRT sample is close to 1 (line 264)
Point 12 (Figure 11):
Scale bar is needed in the fracture surfaces.
Response: done
Point 13 (Discussion, Figure 13):
In the manuscript, the effect of critical hydrogen concentration on hydrogen
embrittlement was discussed. However, if applied stress is changed, the
critical hydrogen content should be changed. The authors should discuss the
hydrogen embrittlement properties with both critical hydrogen content and
applied stress.
Response: some further information has been added
Reviewer 2 Report
The paper presents some interesting issues, but several aspects/modifications should be implemented in order to publish it.
- English should be checked by a native speaker. There are mistakes in prepositions and expressions. A native revisión is mandatory even if everything is understandable.
- In English decimal numbers should be separated by "." instead of "," as it is done. This should be changed. Also the Word "probe" should be changed by sample or specimen.
- Some paragraphs of the text are not justified. Do it.
- Abstract; It says "ultimate tensile strenght higher than 1000MPa", it is not higher than 1400MPa?
- 1. Introduciton; lines 41-43 should be justified by adding some references.
- 1. Introduciton; lines 44-45: HEDE and HELP are mentioned, but there are more micromechanisms. Check and complete this in bibliography.
- 2.1 Materials; The composition, as well as the mechanical properties of the steels are given. It should be necessary to include the ppm of Hydrogen content of the materials in the "as received" state, as hydrogen content determination tests are performed in this work.
- 2.2 Hydrogen charging; lines 83-84: "to avoid Surface damage". It should be explained what kind of Surface damage is avoided and why it could take place.
- 2.2 Hydrogen charging; lines 85-88; 10 minutes of exposition in an oven is a lot of time ot allow hydrogen diffussion out of the sample. Why tests just after charging are not considered. Explains this and, if tests were carried out include this in the paper.
- 2.3 Slow strain rate tests; Why a 0.001mm/s rate was chosen? Is this slow enought? What is the deformation rate in s-1 (1/s) as recommended by standards? Also, why the samples geometry chosen is not a cylindrical bar? Justify this issues.
- 2.4 Four point bending tests; where was this sample geometry taken from? why this geometry was chosen? Justify it.
- 2.4 Four point bending tests; Standard ASTM F-1624 is now well known by the whole scientific comunity. A slight explanation of it principles should be included. Also, why the load was increased up to 90% of Rp0.2 and if maintained during 24 the test is considered safe? If ASTM F-1624 is applied the load is increased up to braking the specimen; this is not clear at all, explain this better or correct it if necessary.
- 3.1 Slow strain rate tests; it is not clear at all if the test takes place exposing the sample to a continuous source of hydrogen after precharging (constant saturated hydrogen content) of if the sample is precharged and then tested in air. Clarify this fact because it is difficult to follow the work. In any case, were hydrogen content test done before and after the test? If so include them.
- Figures 6 and 7 should be enlarged and the numbers Font should be increased, in order to see the numbers in the axes.
- 3.2 Four point vending test; Why the specimens are just charged up to 0.8ppm? Is there a reason? This content should be very close to the "as received" one (this data should be included in table 2) and may not be very representative of embrittlement. Justify this. Why samples were not charged up to 4 or 5ppm? (This can be related with the graph in section 4.1)
- 3.3 SEM images; Figure 11; it should be clarified to which test correspond the imagen from Figure 11.
-3.4 FEM simulation; what is the objective of this section? It is not clear why this is inlcuded? It should be eliminated as its relatioship with the rest of the work is not evident; if it is kept it should be justified very well the target of this section.
- As section 4 just has subsection 4.1 this should be eliminates re-naming section 4 without any subsections.
-5 Conclusions; These conclusions seem to represent results from tests on two certain materials froma a complete characterization than a research work. These conclusions do not give anything else than data comparing two steels. Stronger research conclusions should be issued; a scientific paper should contain more that just tests results. Hydrogen content tests and micrographies could be included in order to enter in the micromechanisms analysis of the test carried out.
Author Response
Response to Reviewer 2 Comments
Point 1:
English should be checked by a native speaker. There are mistakes in
prepositions and expressions. A native revisión is mandatory even if
everything is understandable.
Response: an English revision was done
Point 2:
In English decimal numbers should be separated by "." instead of "," as it is
done. This should be changed. Also the Word "probe" should be changed by
sample or specimen.
Response: Done; probe refers to HELIOS innovative instrument, authors don’t mean sample.
Point 3:
Some paragraphs of the text are not justified. Do it.
Response: Done
Point 4:
Abstract; It says "ultimate tensile strength higher than 1000MPa", it is not
higher than 1400MPa?
Response: the reference to 1000 MPa of ultimate tensile strength is general to indicate high strength steels; in this work, in particular, both the materials have an ultimate strength major than 1400 MPa
Point 5:
Introduction; lines 41-43 should be justified by adding some references.
Response: done
Point 6:
Introduciton; lines 44-45: HEDE and HELP are mentioned, but there are
more micromechanisms. Check and complete this in bibliography.
Response: an other micromechanism is added but the main idea was to named the most famous ones just for completeness.
Point 7:
Materials; The composition, as well as the mechanical properties of the
steels are given. It should be necessary to include the ppm of Hydrogen
content of the materials in the "as received" state, as hydrogen content
determination tests are performed in this work.
Response: the diffusible hydrogen content in as received material is 0 ppm
Point 8:
Hydrogen charging; lines 83-84: "to avoid Surface damage". It should be
explained what kind of Surface damage is avoided and why it could take
place.
Response: to avoid surface damage means that, due to the local absorption of hydrogen, it is possible formation of surface cracks.
Point 9:
Hydrogen charging; lines 85-88; 10 minutes of exposition in an oven is a
lot of time ot allow hydrogen diffussion out of the sample. Why tests just after
charging are not considered. Explains this and, if tests were carried out
include this in the paper.
Response: 10 minutes of exposition in an oven is necessary to simulate paint baking process (a real manufacturing process) and it’s useful to homogenize the hydrogen concentration in the material; the release of hydrogen is negligible since the material is Al-Si coated.
Point 10:
2.3 Slow strain rate tests; Why a 0.001mm/s rate was chosen? Is this slow
enought? What is the deformation rate in s-1 (1/s) as recommended by
standards? Also, why the samples geometry chosen is not a cylindrical bar?
Justify this issues.
Response: the crossbar speed of 0.001 mm/s corresponds to a deformation measured by means of an extensometer of 4.5 E-5 s-1, inside the range recommended by standards.
The geometry is not cylindrical because samples have to simulate sheet products of automotive industry.
Point 11:
Four point bending tests; where was this sample geometry taken from?
why this geometry was chosen? Justify it.
Response: the geometry of 4PB samples was chosen in order to study the behaviour of flat products; moreover the hole is added because a lot of fractures occur starting from holes in the real sheets in automotive application.
Point 12:
Four point bending tests; Standard ASTM F-1624 is now well known by
the whole scientific comunity. A slight explanation of it principles should be
included. Also, why the load was increased up to 90% of Rp0.2 and if
maintained during 24 the test is considered safe? If ASTM F-1624 is applied
the load is increased up to braking the specimen; this is not clear at all,
explain this better or correct it if necessary.
Response: it was chosen a limit of 90% of Rp0.2 because this kind of material is always used in elastic range
Point 13:
Slow strain rate tests; it is not clear at all if the test takes place exposing
the sample to a continuous source of hydrogen after precharging (constant
saturated hydrogen content) of if the sample is precharged and then tested in
air. Clarify this fact because it is difficult to follow the work. In any case, were
hydrogen content test done before and after the test? If so include them.
Response: the measurements of diffusible hydrogen were performed just after the end of the test.
Please note the test duration is always few minutes and the samples are silicon-aluminized (coating acts as a barrier for hydrogen)
Point 14:
Figures 6 and 7 should be enlarged and the numbers Font should be
increased, in order to see the numbers in the axes.
Response: done
Point 15:
Four point vending test; Why the specimens are just charged up to
0.8ppm? Is there a reason? This content should be very close to the "as
received" one (this data should be included in table 2) and may not be very
representative of embrittlement. Justify this. Why samples were not charged
up to 4 or 5ppm? (This can be related with the graph in section 4.1)
Response: the diffusible hydrogen for this type of steels in as received condition is close to zero; moreover reaching a diffusible hydrogen content of 4 or 5 ppm is not necessary because, as it can be seen in the mechanical tests, the threshold value is always lower than 1 ppm.
Point 16:
SEM images; Figure 11; it should be clarified to which test correspond
the imagen from Figure 11.
Response: done
Point 17:
FEM simulation; what is the objective of this section? It is not clear why
this is inlcuded? It should be eliminated as its relatioship with the rest of the
work is not evident; if it is kept it should be justified very well the target of this
section.
Response: FEM simulation has been deleted
Point 18:
As section 4 just has subsection 4.1 this should be eliminates re-naming
section 4 without any subsections.
Response: subsection 4.1 has been deleted
Point 19:
Conclusions; These conclusions seem to represent results from tests on
two certain materials froma a complete characterization than a research work.
These conclusions do not give anything else than data comparing two steels.
Stronger research conclusions should be issued; a scientific paper should
contain more that just tests results. Hydrogen content tests and
micrographies could be included in order to enter in the micromechanisms
analysis of the test carried out.
Response: we applied a scientific approach to study hydrogen embrittlement of advanced high strength steels carrying out a large series of tests to compare different mechanical characterizations and to identify the best, in a very practical way.
Investigation of micromechanisms wasn’t the aim of this work.
Reviewer 3 Report
In this article the authors propose an experimental methodology to characterize the hydrogen embrittlement of ultra high strength steels.
The following improvements are proposed:
1. The introduction needs more bibliographic support.
2. Once the problem has been identified and the mechanisms of embrittlement (line 50) are stated, an introduction to the proposed experimental methodology must be made.
3. It is necessary to show in a flowchart the different stages that are followed in the experimental methodology. Due to the high experimental load, all the stages followed must be explained and justified in order to identify the results obtained more clearly. This is essential. On the contrary, it is difficult to understand the interpretation of the results obtained
4. "Table 1" is not named in the text before it appears.
5. The geometry of the specimens must be justified.
6. The quality of figures 6,7,8 and 9 should be improved. The format of the journal should be followed, figure caption 8 and 9.
7. The results should be commented and discussed at the same time. In this way, problems like "line 158" would be avoided. In this line it talks about the m3 parameter which appears explained on line 263. In addition everything would be understood much better.
8. In Figures 6 and 7, what does the threshold value represent?
9. Although this is not the object of the study, in the tensile tests of the preloaded samples, why don't you study the evolution of stress/strain with respect to hydrogen embrittlement?
10. The SEM study must be completed with a composition study (EDS). Line 191. The equipment used must be named in the experimental methodology.
11. As it is explained, the FEM study does not contribute to the article.
Author Response
Response to Reviewer 3 Comments
Point 1:
The introduction needs more bibliographic support
Response: some additional references are added
Point 2:
Once the problem has been identified and the mechanisms of
embrittlement (line 50) are stated, an introduction to the proposed
experimental methodology must be made.
Response: done
Point 3:
It is necessary to show in a flowchart the different stages that are
followed in the experimental methodology. Due to the high experimental load,
all the stages followed must be explained and justified in order to identify the
results obtained more clearly. This is essential. On the contrary, it is difficult
to understand the interpretation of the results obtained
Response: Answering to reviewer number 2, test procedure was clarified in the test (probably now a flowchart is not necessary)
Point 4:
Table 1" is not named in the text before it appears.
Response: Now it’s named
Point 5:
The geometry of the specimens must be justified.
Response: the geometry (SSR samples) refers to SEP 1970:2011 (reference standard), while the 4PB geometry has been chosen to simulate vehicle components which typical fracture starts from holes
Point 6:
The quality of figures 6,7,8 and 9 should be improved. The format of the
journal should be followed, figure caption 8 and 9.
Response: done
Point 7:
The results should be commented and discussed at the same time. In
this way, problems like "line 158" would be avoided. In this line it talks about
the m3 parameter which appears explained on line 263. In addition everything
would be understood much better.
Response: According to traditional literature, results are shown and discussed in two different paragraphs; however, in the previous line 158, reference to m3 is deleted.
Point 8:
In Figures 6 and 7, what does the threshold value represent?
Response: the meaning of threshold value is added.
Point 9:
Although this is not the object of the study, in the tensile tests of the
preloaded samples, why don't you study the evolution of stress/strain with
respect to hydrogen embrittlement?
Response: because the sample is not the standard tensile test.
Point 10:
The SEM study must be completed with a composition study (EDS). Line
191. The equipment used must be named in the experimental methodology.
Response: EDS microanalysis was not considered because fracture surface aspect is very clear and no effect of inclusions or precipitations are visible; however EDS analysis was performed on USIBOR 1500 [Valentini, Tedesco, Corsinovi, Bacchi]
Point 11:
As it is explained, the FEM study does not contribute to the article.
Response: FEM simulation has been deleted.
Round 2
Reviewer 1 Report
There are some spelling mistakes in the text, and authors should improve English. Please check the manuscript before resubmitting. Questions and comments are as follow. Please check and consider the manuscript.
(1) Introduction, line 26-27:
What is “high-grade”? Reader cannot understand the meaning.
(2) line 149:
Was the NH4SCN content in mass% or atomic%?
(3) line 152-156, line 258-260:
In point 4 in your response, you explain that hydrogen was not desorbed during bake hardening process because of Si and Al layer. Assuming that, hydrogen cannot be absorbed during hydrogen charging, and measurement of hydrogen content at 265 ℃ cannot be conducted because Si and Al layer protect hydrogen absorption and desorption.
(4) SSRT
In point 5 in your response, crosshead speed of 0.001 mm/s corresponds to strain rate of 4.5x10-5 /s. From the crosshead speed and strain rate, the gauge length is calculated as 22.2 mm. However, SSRT sample has a hole of 20 mm in diameter. It is expected that deformed part in the SSRT sample was only center of specimen. Therefore, strain rate which you estimated was incorrect. Strain rate may be high in this experiment.
(5) line 358:
The information of equipment of SEM should explain in experimental procedure.
(6) line 523:
I could not understand the conclusion. What was “concentration”, “stress concentration” or “hydrogen concentration”? Authors should explain more politely.
Author Response
Response to Reviewer 1:
1) With High grade materials authors mean High Strength Steels with particular reference to Automotive steels (so Ultimate tensile strength of 1500 or 2000 MPa were considered in the text).
2) It's the Mass %
3)It should be considered that at room temperature the Al-Si coating acts as a barrier, thus hydrogen charging happens mainly through the edges, and this was one of the reasons to perform paint baking simulation process so homogenize diffusible hydrogen content inside the samples.
Moreover, once the sample is heated at higher temperature diffusion is enhanced through the coating. At paint baking temperature and considering the small duration of the process, hydrogen desorption through the coating can be still considered negligible, while at 265°C all diffusible hydrogen desorbs from the sample (Ref. [13])
4) Since the tested specimen presented a hole at the center, the strain rate is not uniform along the specimen itself.
Considering a conventional, not notched, flat specimen with gauge length of 1 inch, a crossbar speed of 0.001 mm/s corresponds to a strain rate of 4x10-5 s-1 which is in agreement with the standard. Thus, it was assumed acceptable to use the same speed for these experiments.
5) they have been reported in "Materials and methods" section
6) Stress "concentration" has been replaced by stress "intensification" to differentiate from hydrogen "concentration"
Reviewer 2 Report
The comments were addressed.
Author Response
No reply needed
Reviewer 3 Report
Although authors have made some of the proposed changes, the article still looks like a technical report and it is hard to analyze the discussion of the results obtained. In this sense, the discussion should be deeply, but not only with a phenomenon description.
Author Response
No reply needed